Advanced neural network-based model for predicting court decisions on child custody

Abrar Mohammad 1
Salam Abdu 2
Ullah Faizan 3
Nadeem Muhammad 4
AlSalman Hussain halsalman@ksu.edu.sa 5
Mukred Muaadh 6
Amin Farhan farhanamin10@hotmail.com 7
1 Faculty of Computer Studies, Arab Open University , Muscat , Oman
2 Department of Computer Science, Abdul Wali Khan University , Mardan , Pakistan
3 Department of Computer Science, Bacha Khan University , Charsadda , Pakistan
4 Department of Computer Science and Software Engineering, International Islamic University, Islamabad , Islamabad , Pakistan
5 Department of Computer Science, King Saud University , Riyadh , Saudi Arabia
6 Department of Business Analytics, Sunway University , Selangor , Malaysia
7 School of Computer Science and Engineering, Yeungnam University , Gyeongsan , South Korea
Ali Hazrat
Electronic publication date: 2024 Oct 22
Publication date: 2024
Volume: 10
Electronic Location ID: e2293
Received 2024 Apr 29; Accepted 2024 Aug 9
Copyright: ©2024 Abrar et al.
Copyright year: 2024
Copyright holder: Abrar et al.
License: This is an open access article distributed under the terms of the Creative Commons Attribution License, which permits unrestricted use, distribution, reproduction and adaptation in any medium and for any purpose provided that it is properly attributed. For attribution, the original author(s), title, publication source (PeerJ Computer Science) and either DOI or URL of the article must be cited.
License URL: https://creativecommons.org/licenses/by/4.0/

Keywords: Neural networks, Natural language processing, Criminal law, Family law, Legal informatics, Litigation prediction, Legal outcome forecast, Predictive algorithms, Machine learning, Artificial intelligence

Funding: The Researchers Supporting Project, King Saud University, Riyadh, Saudi Arabia RSP2024R244 This work was supported by the Researchers Supporting Project, King Saud University, Riyadh, Saudi Arabia, under Grant RSP2024R244. The funders had no role in study design, data collection and analysis, decision to publish, or preparation of the manuscript.

==============================
Predicting court rulings has gained attention over the past years. The court rulings are among the most important documents in all legal systems, profoundly impacting the lives of the children in case of divorce or separation. It is evident from literature that Natural language processing (NLP) and machine learning (ML) are widely used in the prediction of court rulings. In general, the court decisions comprise several pages and require a lot of space. In addition, extracting valuable information and predicting legal decisions task is difficult. Moreover, the legal system’s complexity and massive litigation make this problem more serious. Thus to solve this issue, we propose a new neural network-based model for predicting court decisions on child custody. Our proposed model efficiently performs an efficient search from a massive court decisions database and accurately identifies specific ones that especially deal with copyright claims. More specially, our proposed model performs a careful analysis of court decisions, especially on child custody, and pinpoints the plaintiff’s custody request, the court’s ruling, and the pivotal arguments. The working mechanism of our proposed model is performed in two phases. In the first phase, the isolation of pertinent sentences within the court ruling encapsulates the essence of the proceedings performed. In the second phase, these documents were annotated independently by using two legal professionals. In this phase, NLP and transformer-based models were employed and thus processed 3,000 annotated court rulings. We have used a massive dataset for the training and refining of our proposed model. The novelty of the proposed model is the integration of bidirectional encoder representations from transformers (BERT) and bidirectional long short-term memory (Bi_LSTM). The traditional methods are primarily based on support vector machines (SVM), and logistic regression. We have performed a comparison with the state-of-the-art model. The efficient results indicate that our proposed model efficiently navigates the complex terrain of legal language and court decision structures. The efficiency of the proposed model is measured in terms of the F1 score. The achieved results show that scores range from 0.66 to 0.93 and Kappa indices from 0.57 to 0.80 across the board. The performance is achieved at times surpassing the inter-annotator agreement, underscoring the model’s adeptness at extracting and understanding nuanced legal concepts. The efficient results proved the potential of the proposed neural network model, particularly those based on transformers, to effectively discern and categorize key elements within legal texts, even amidst the intricacies of judicial language and the layered complexity of appellate rulings.

Introduction

The child custody decision after a divorce or separation is a serious issue and often arises in a series of legal cases (Medvedeva, Vols & Wieling, 2020). According to recent research, Europe had an estimated 800,000 divorce cases in 2020 (Campisi et al., 2020), while the United States had 630,505 divorces and a whopping 51.1% of marriages (Aletras et al., 2016). Another analysis of judicial decisions emanating from Spanish courts in 2021 was 86,851. The 21.2% escalated to courtroom litigations, with minors involved in 53.2% of the disputes. In addition, custodial assignments were in favor of mothers (53.1%), while fathers were granted custody in merely 3.5% of the cases, and joint custody arrangements were made in 43.1% (Cuchillo Pocco, 2021). For scholars and practitioners within the realms of legal sociology, legislation, and public policy, acquiring a deep understanding of judicial decision-making processes in child custody matters is a common interest. Despite the statistical data provided, there is a strong need to look at more up-to-date statistics, given the evolving nature of family legal disputes. In this context, we believe that artificial intelligence (AI), especially through the use of linguistics plays an important role in the computer analysis of court docket decisions. By extracting and synthesizing information from judgments, parental requests, judicial decisions, and original reasons. The AI provides useful information (Hoadley & Lucas, 2018). Natural language processing (NLP) stands at the forefront of AI applications in the legal field, expanding its application beyond the domain of entity recognition to include content mining, thereby enabling the identification of complex entities and content patterns within the legal literature (Park et al., 2020). Research in this field often explores the application of machine learning techniques to legal texts with a focus on outcome prediction, which lacks an in-depth analysis of legal reasoning or argumentation structure. Previous research primarily suffered from a narrow scope, only addressing simple classification problems without capturing the intricate details of legal arguments and decisions. The transformer-based models were previously applied to child custody prediction. However, the novelty of this research is the integration of bidirectional encoder representations from transformers (BERT) and bidirectional long short-term memory (Bi_LSTM). These suggestions make a more nuanced, context-rich analysis of court rulings. Our proposed dual-stage approach enhances the understanding of legal reasoning by performing both binary and multilabel classifications of legal texts and individual sentences, capturing the complexities of judicial arguments more effectively. This proposed method provides a deeper, more detailed insight into judicial decisions, potentially offering enhanced legal analysis tools that surpass existing methodologies’ capabilities. This advancement is critical for enhancing legal research, practice, and education by enabling a more nuanced examination of court rulings. Our research uses NLP techniques to classify and categorize the main elements of court decisions, such as plaintiffs’ requests, judicial decisions, and legal arguments used, to make 3,047 judicial decisions. Our proposed model is trained and tested using a neural network model. The objective is to mimic the analytical skills of human commentators (Watson et al., 2022). In this research, we introduce a new application of NLP to analyze judicial decisions. we propose a two-dimensional analysis model, which includes two sets of broad texts and several sets of independent sentences (McKnight, Kacmar & Choudhury, 2004). This two-stage process is particularly adept at identifying relevant issues from long and complex legal document reports, which often include citations and reports of earlier. Our research highlights the agreement between human interpreters and an AI model in interpreting legal concepts. The novelty of the proposed model is the integration of BERT and Bi-LSTM. The traditional methods are primarily based on support vector machines (SVM) and logistic regression. We have performed a careful analysis of a large dataset containing more than 3,000 court judgments (Price, 2015). Briefly, our research proposes a new tool that can search a collection of court decisions to identify those that deal with copyright claims. The proposed tool’s primary function is to extract pertinent data from these decisions to address inquiries such as the frequency of courts aligning with the custody form proposed by the plaintiff, whether the likelihood of agreement varies between individual and joint custody requests, and the prevalence of specific arguments in the justification of either form of custody. The key objectives of our research are:

• Herein, we propose a neural network model for classifying sentences in custody-related court decisions, i.e., distinguishing between case rulings and legal precedents.

• The proposed models can identify key components of court rulings, for instance, plaintiff’s requests, court decisions, and critical arguments.

• The efficiency of the proposed model is measured by deriving the court’s rationale from the judgment texts to understand the underlying legal reasoning without needing external context.

• The analysis of child custody-related court rulings through the application of advanced neural network models, with a specific focus on achieving optimal F1 scores.

The key contributions of our research are given below:

1. In this research, we introduce an innovative application of NLP to comprehensively analyze entire judicial rulings. Our proposed analytical approach involves the binary classification of extensive texts and the multi-classification of individual sentences. Our proposed dual-stage model is particularly adept at discerning relevant content from the often lengthy and complex narratives typical of legal documents.

2. The novelty of the proposed model is the integration of BERT and Bi-LSTM. The traditional methods are primarily based on SVM, and logistic regression.

3. Our research highlights the consensus between human annotators and the AI model from legal aspects.

The rest of this article is structured as follows: Section ‘Literature Review’ introduces related work, offering insights into comparable studies previously conducted. We then proceed with a comprehensive explanation of our methodology, including the annotation process and the strategies employed for model training, detailed in Section ‘Methodology’. Section ‘Experimental Results’ is dedicated to showcasing our findings, followed by an in-depth discussion in Section ‘Discussions’ that covers both the consensus among annotators and the neural network’s performance, especially its alignment with human judgment. We conclude the article with our final thoughts and conclusions in Section ‘Conclusion’.

Literature Review

Advanced techniques in legal information extraction through NLP

The NLP uses in legal texts facilitates the extraction of different information, an important task at the intersection of NLP and legal studies. Named entity recognition (NER) is the basic method in this area, which is tasked with the identification of different legal entities such as individuals, legal professionals, geographical areas, corporate bodies, legal documents, and others (Correia, 2018). This process extends to recognizing people’s specific roles during litigation (Gupta et al., 2018) and establishing links between specific legal entities. NER methods are distinguished from knowledge-driven systems, which leverage domain-specific knowledge, lexicons, and combinations of linguistic features and contexts as well as learning algorithms, and neural network systems that minimize feature engineering, they often use word and form-level details to adapt to the nuances of the text (Mohapatra, 2018). Recent advances have visible the integration of advanced textual content processing techniques, especially word processing, to noticeably enhance the performance of neural networks in NLP obligations (Krasadakis, Sakkopoulos & Verykios, 2024). Argument mining, another crucial issue, specializes in identifying and analyzing arguments in felony texts, highlighting the usage of legal norms and precedents which might be often combined with quotations (Savelka, 2020). This consists of the vast paintings of citation mining, which categorizes citations primarily based on their referential feature in the felony language (Lawrence & Reed, 2020). Scharpf et al. (2023) embarked on a similar task, using machine learning to automatically discover principles and legal facts within citations, recognizing the distinctiveness of such features and the difference in the analysis of text among different reviewers. Their work, which reflects the aims and methods of our study, achieved inter-annotator agreement (IAA) scores, as measured by Cohen’s Kappa, of K = 0.65 for IAA and K = 0.72 for the reporter’s concordance. With the expansion of the field, many studies have delved into obtaining concrete data from legal texts, such as Ghimire, Kim & Acharya (2023), which extracted information on evidence from Chinese court documents, used a dual-task approach that included classifying and analyzing sentences of evidence construction and their task yielded an F1 score of it was 0.72, which demonstrated the efficiency of the method. In the area of predictive justice, there is a growing interest in using these NLP techniques to predict judicial outcomes. Importantly, family law systems remain weak, with notable exceptions such as Zeleznikow (2021), which uses network and rule-based reasoning for property division estimates in Australian divorce cases. Similarly, Gepp (2015) and Goyal, Pandey & Jain (2018) used Markov networks and CHAID analysis, respectively, to predict the distribution of rights in Chinese courts and Taiwanese courts, demonstrating the ability of these methods to significantly influence predicted justice in family law with F1 scores as high as 0.9783. This exploration of NLP in legal text analysis not only underscores the depth and breadth of existing methods but also highlights expansive possibilities for future research and application, particularly in enhancing the understanding and prediction of legal outcomes.

NLP and deep learning in child custody prediction

Recent advancements in NLP and deep learning (DL) have significantly impacted the field of legal decision prediction, including child custody cases. Several studies have explored the use of NLP techniques to automate the analysis of legal texts and predict judicial outcomes. For instance, Liang (2021) utilized deep learning models to analyze textual data from child custody rulings, demonstrating how these methods can predict case outcomes with high accuracy by extracting and learning from patterns in judicial language and reasoning. Liang (2021) applied NLP methodologies to the European Court of Human Rights decisions, highlighting the potential of machine learning for understanding legal arguments and decision-making processes in the context of custody disputes. In child custody prediction, NLP applications often involve tasks such as entity recognition, argument mining, and sentiment analysis to identify critical factors influencing judicial decisions. These techniques are complemented by DL models that can handle complex, context-rich legal texts, enabling more precise predictions. For example, Liang (2021) explored machine learning approaches to predict judicial outcomes in European Court of Human Rights cases, providing insights into how similar techniques can be adapted for child custody predictions by learning from historical cases and the specific language used in custody rulings. The NLP landscape has changed dramatically with the advent of deep learning technologies, leading to advances in various fields including entity recognition, text classification, machine translation, querying, and language generation (Goyal, Pandey & Jain, 2018; Wu 2020). Among the examples used, the transformer architectures introduced by Chitty-Venkata et al. (2023), emerged as the cornerstone, underpinning ELMO, BERT, GPT series, RoBERTa, and other models, and advanced most of the current research in this area (Su, 2024). Transformers excel because they rely on cognitive methods, enabling them to develop comprehensive language models that can generalize and contextualize representations This study uses the BERT architecture by Google (Devlin et al., 2019) distinguished by two-way training and based on context and the ability to impose contextual semantics, which is a huge leap above models such as Word2Vec or GloVe which are seamlessly loaded. Despite its potential, Transformer models, including BERT, face challenges such as fixed input length, word input issues, and computational requirements (Raiaan, 2024). Proposals such as XLNet, RoBERTa, and DistilBERT have been developed to address these limitations, each providing significant improvements in model learning and efficiency. GPT-3′s ability to generate highly coherent and contextually relevant text surpasses traditional models like BERT. Its advanced understanding of complex language structures can improve the model’s accuracy in interpreting and predicting legal decisions, especially in handling nuanced legal arguments and diverse linguistic patterns found in court rulings. GPT-3′s capacity to process extensive contexts can be particularly advantageous in the legal domain, where understanding the subtleties of legal language and the relationships between various sections of court documents is crucial. This feature enables more precise extraction of key elements from legal texts, such as judicial reasoning and argumentation patterns. Legal-BERT, a variant tailored for legal texts, underscores the nuanced application of BERT in domain-specific contexts, suggesting that conventional pretraining and finetuning strategies may not directly translate to specialized fields such as law (Lothritz et al., 2020). This highlights the adaptability required in applying Transformer models to domain-specific NLP tasks, ranging from text classification to argument mining. Transformers have also paved the way for innovative applications in the legal domain, including generating visual summaries of court rulings and assisting in argument mining within legal texts (Lothritz et al., 2020). The emergence of legal assistants like LawGPT showcases the potential of Transformer models to revolutionize legal research and practice. A variant designed for legal texts i.e., Legal-BERT, highlights the different uses of BERT in specific contexts, suggesting that common training and correction strategies may not directly translate to features. specific as a rule (Nityasya et al., 2022). This highlights the flexibility required when applying Transformer techniques to specific NLP tasks, from text classification to argument mining. Transformers have also opened the way for new applications in the legal field, including generating visual summaries of court judgments and assisting in mining legal documents (Curello, 2023). The emergence of legal assistants such as LawGPT demonstrates the potential of Transformer models to transform legal research and practice (Cyphert, 2021). The transformative impact of transformer models in NLP, particularly within specialized domains such as law, represents a significant evolution in the ability to process, understand, and generate human language, ushering in a new era of AI-powered legal analysis and assistance.

Methodology

This section delves into the cutting-edge methods of extracting pertinent information from legal documents using NLP. The focus here is on the nuanced application of NLP techniques, especially in the realm of legal studies, where the identification of entities and the extraction of arguments from legal texts are paramount. By employing advanced NLP technologies, the study aims to enhance the precision and depth of legal text analysis, shedding light on the intricate dynamics of legal reasoning and the utilization of legal norms and precedents. The proposed methodology for analyzing custody-related court rulings using neural networks is depicted in Fig. 1.

Figure 1 Proposed methodology for analyzing custody-related court rulings using neural networks.

Criteria for selecting annotation categories

In developing the framework for our study, an important step involved identifying specific aspects of judicial decisions that should be clarified. This process was guided by three important factors: (1) maintaining a manageable number of groups to ensure the success of the model; (2) selecting groups that were sufficiently common across the dataset to be statistically significant; and (3) aligning the designated groups with the legal standards that govern copyright decisions. The completed sections were organized into three different groups: the nature of the appeal and the court’s judgment, the legal principles used, and the evidence presented.

Categorizing custody requests and judicial decisions

The main focus of our application process was the classification of claims related to the obligations made by the plaintiff and the subsequent decisions by the jury.

A key distinction was drawn between two types of physical custody: sole custody, where the child lives with one parent while the other is granted visitation rights, and shared custody, characterized by the exchange of the child’s residence between both parents at approximately equal intervals. These are defined as RQ_JOIN (for requests) and DEC_JOIN (for court decisions), with the addition of a “+” sign to indicate a request or a copyright decision and a “–” sign for a shared right.

Prioritizing the child’s best interests in legal judgments

In the analysis of this research the primary legal doctrine guiding child custody decisions was the best interests of the child, overriding other potential principles such as parental equality due to data is not readily available. This basic principle of family law mandates that the welfare of the child takes precedence over the wishes or rights of the parents. Despite its importance, the application of this principle can be ambiguous, allowing for different interpretations that can support different care arrangements (Rawls, 1997). This issue is defined in our database using the BEST_INT label. The complex structure of family law in Spain, characterized by regional differences in law, requires a clear analysis of the factual factors used by judges in custody decisions (Hayden, 2011). Our comprehensive review of these laws has led to the selection of key factors considered in judicial decisions, which we have broken down for analysis as summarized in Table 1. The analysis aim of this research is to first identify the nature of custody requests and ascertain whether these were upheld by the judiciary, ensuring consistency in the annotation of these primary requests and decisions. The secondary goal was to document the rationale underpinning judicial decisions, distinguishing between legal principles, particularly the child’s best interests, and the factual arguments outlined above. This approach enables an exploration of the prevalence and influence of these arguments in judicial reasoning, offering insights into potential correlations with the outcomes of custody cases and the demographics of the involved parties.

Table 1 Categories of factual arguments in custody analysis.

Category	Description	
CHILD_CIRC	Pertains to the child’s context, including age, health, educational performance, and any special needs.	
CHILD_ROOT	The child’s connections to their community, school, or a particular parent.	
CHILD_OPIN	The preferences or views of the child regarding custody arrangements.	
PAR_RELAT	The nature of the relationship between the parents and its impact on the child’s welfare.	
PAR_RDNS	Each parent’s capacity to provide for the child, including time availability and financial resources.	
PAR_DED	The level of care and involvement each parent has historically demonstrated towards the child.	
PSY_REP	Insights from psychological and social evaluations conducted by experts.	

Structure and analysis of judicial decisions

Spanish judicial decisions are structured into four distinct segments. Initially, the heading provides an overview, listing the court, involved parties, and their legal representatives. The subsequent facts section outlines the procedural history, claims, and evidence presented by the parties. This is particularly detailed in appellate decisions, which recapitulate the initial proceedings, including the claims made and the lower court’s judgment. The legal grounds segment follows, presenting the rationale behind the court’s decision, the ratio decidendi, defined by Raz (2002) as the legal principles derived from the facts that underpin the ruling. This section may also address procedural matters and ancillary issues. The concluding part of the decision, the verdict, succinctly states the court’s determination. For our research, identifying the most informative sections of the decision for analysis was crucial. The verdict explicitly states the outcome concerning the initial request, indicating acceptance, partial acceptance, or denial, and is typically straightforward to identify using text analysis tools. However, without knowledge of the original request, this information is incomplete, not specifying the nature of the custody granted. The facts section, while comprehensive, often includes extraneous information, rendering it less efficient for analysis. In contrast, the legal grounds section not only clarifies the appellate claims and the court’s stance on them but also filters the relevant facts for constructing the decision’s justification, incorporating the applicable legal standards and principles.

Figure 2 shows the structural elements of court rulings and the categorization schema applied in their analysis. It details the specific segments of legal decisions scrutinized and categorizes them according to the analytical methodology employed in the study. Our hypothesis posited that the legal grounds alone might suffice to extract all required elements for our study, leading to the decision to focus our annotation and analysis on this segment. However, distinguishing the core elements of the ratio decidendi within the legal grounds posed challenges, as courts often recount the arguments of both parties, cite precedents, and summarize prior proceedings, incorporating language that mirrors those used in the decisive legal reasoning. Thus, a contextual analysis was necessary to accurately label the relevant categories within the legal grounds.

Figure 2 Court ruling structure and analyzed categories.

Compilation and annotation of the appellate court rulings dataset

In the Spanish judicial system, family law cases are initially adjudicated by lower courts, with subsequent appeals being resolved by one of the 50 provincial courts spread across the country. Our dataset was sourced from the Spanish Centre for Judicial Documentation (CENDOJ) (Rawls, 1997), which predominantly archives decisions from provincial and higher courts. Consequently, our analysis was based on appellate decisions, although first-instance rulings would have ideally been more aligned with our research objectives. A notable challenge in dataset compilation was the CENDOJ’s limitation on bulk access to rulings, necessitating manual download of individual documents, which restricted the comprehensiveness of our dataset to include all custody-related rulings within the selected timeframe. The dataset contains 3,047 appellate court decisions focusing on custody arrangements, child support, and division of the family home. Efforts were made to represent all provincial courts in proportion to their population size, with a preference for more recent cases from 2015 to 2020, although the dataset extends back to 2006, reflecting the legal recognition of joint custody in Spain. is This collection accessible for further research at Munoz Soro & Serrano-Cinca (2021) also provides extensive information on the dataset’s geographic and temporal distribution, labeling schema, and annotation method. The verdicts, initially in PDF format, were extracted with metadata and text translation, with a rules-based approach that identifies the culprits from the document titles. The main text, especially the legal reasoning part equal to 2,067 words, was annotated by two legal experts using the Brat annotation tool (Stenetorp et al., 2012). This careful process lasted for 10 months, and the annotators received a daily expression of 15 sentences. An annotation guide, developed through the first group discussion and IAA), led this effort, to ensure clarity and consistency in registration. The observations served two purposes: to generate the judgment process and to provide training data for the neural network. The selection process of the text fragments aimed at clearly reflecting the written groups, which do not require additional content, resulting in different volumes of 87 words in length. Challenges arose when multiple levels of the group appeared in one decision. The protocol mandated that a representative section be written for each section, ensuring that repeated labels reflected the consideration of the many aspects of the dispute by the court. Table 2 summarizes the various factors and considerations involved in annotations of texts from court decisions, ranging from custody requests and decisions to psychological evaluations and the child’s social and family environment.

Table 2 Annotated texts from judicial decisions.

Category	Annotated example	
Sole Custody Request (RQ_JOIN +)	The mother contests the custody setup, advocating for sole custody to be awarded to her.	
Joint Custody Request (RQ_JOIN -)	The father seeks an amendment for joint custody, detailing the proposed arrangements present in the legal documents.	
Sole Custody Decision (DEC_JOIN +)	The ruling revokes a previous shared custody order, endorsing sole guardianship.	
Joint Custody Decision (DEC_JOIN -)	The judgment supports reinstating the child’s earlier arrangement of joint custody.	
Child’s Best Interests (BEST_INT)	The court prioritizes the child’s best interests above the parents’ desires in its custody decision.	
Child’s Circumstances (CHILD_CIRC)	The children’s psychological conditions and their ongoing treatments highlight their unique vulnerabilities.	
Child’s Social Connections (CHILD_ROOT)	The eldest child’s preference to maintain his social life in his current environment is acknowledged.	
Child’s Custody Preference (CHILD_OPIN)	The daughter expresses her discomfort with joint custody, preferring to live primarily with her mother.	
Parental Conflict (PAR_RELAT)	The record of mutual complaints between the parents illustrates their profound discord and inability to agree on basic matters.	
Parental Capacity and Support (PAR_RDNS)	The mother’s dependence on her parents for support in child-rearing is noted, indicating a collaborative family support system.	
Parental Involvement Before Separation (PAR_DED)	The mother’s significant role in the child’s care before the separation is highlighted, demonstrating her dedication.	
Expert Psychological Evaluation (PSY_REP)	A professional recommendation suggests that the child’s primary residence be with the mother, with a substantial visitation plan for the father.	

In our research, we focused on a select group of 2,394 judicial decisions, which were primarily centered on child custody issues, although some rulings also addressed child support and the allocation of family residences. In these decisions, a total of 36,087 annotations were identified, an average of 15.07 annotations per order. The distribution of annotations into different categories by the two annotators and their combined totals are detailed in Table 3. The frequency and distribution of the comments made by the two commenters provide information on the different considerations taken in judicial decisions related to child custody in the database.

Table 3 Distribution of annotations across categories.

Category	Annotator 1	Annotator 2	Combined total	
Parental Resources (PAR_RDNS)	4,415	4,087	8,502	
Parental Relationship (PAR_RELAT)	4,616	2,797	7,413	
Child’s Best Interest (BEST_INT)	1,245	2,011	3,256	
Decision for Sole Custody (DEC_JOIN +)	1,498	1,384	2,882	
Request for Joint Custody (RQ_JOIN -)	1,525	1,246	2,771	
Child’s Circumstances (CHILD_CIRC)	1,785	969	2,754	
Request for Sole Custody (RQ_JOIN +)	1,097	852	1,949	
Decision for Joint Custody (DEC_JOIN -)	925	900	1,825	
Child’s Community Ties (CHILD_ROOT)	830	462	1,292	
Child’s Custody Preference (CHILD_OPIN)	628	604	1,232	
Parental Dedication (PAR_DED)	753	478	1,231	
Psychological Report (PSY_REP)	541	439	980	
Total Annotations	19,858	16,229	36,087	

Data annotation process

The annotation process was integral to preparing our dataset for training the NLP model. Here’s an overview of how this was conducted. Annotators were selected based on their proficiency in Spanish and their understanding of legal terminology relevant to child custody cases. Our team included native speakers with backgrounds in law or linguistics. A standardized guideline was developed to ensure consistent annotation across the dataset. This guideline included definitions of key elements such as custody requests, judicial decisions, and significant arguments. Annotators were trained on these guidelines to minimize discrepancies. Annotators reviewed each document, tagging relevant sentences and sections according to the guidelines. This included marking phrases that identified the plaintiff’s requests, court decisions, and arguments used by judges. To ensure accuracy, each annotated document was reviewed by a second annotator. Discrepancies were resolved through discussion or by consulting a senior reviewer. This validation step ensured high inter-annotator agreement and reliable annotations.

The annotated data were used to train and validate our NLP model, providing the basis for its ability to identify and analyze legal text effectively. The high-quality annotations were crucial for achieving the model’s performance metrics.

Developing the classification model

Our aim is to identify different categories of law that can be interpreted in several ways within the court literature. For example, references to the parents’ financial situation can vary greatly, from direct statements about their financial ability to various discussions about their stability at work and dependence on family support because of these variables, a simple rule-based approach was deemed inadequate. Instead, we used a language method based on transformers to improve its ability to understand semantics, which allows recognition of these groups despite their different expressions. Figure 3 depicts the method used to classify sentences within legal texts, separating them based on whether they have been annotated, with a focus on the context that determines this status. The first step in editing involved breaking down the text into single sentences, a task made difficult by legal writing’s propensity for long sentences and typographical errors. Sentences with fewer than three words or more than 300 words were excluded due to the possibility of being uninformative or incorrectly punctuated respectively. The remaining 72,261 sentences were divided into training (72%), testing (18%), and validation groups (10%).

Figure 3 Classification of sentences distinguished by contextual annotation Status.

Figure 4 illustrates the application of multilabel classification leveraging transformer architectures in the analysis of legal documents, with sentences lacking annotations explicitly marked by ‘X’.

Figure 4 Multilabel classification scheme utilizing transformer models (sentences marked with ‘X’ indicate non-annotation).

The classification framework consisted of two main phases. The first phase involved binary categories to determine the presence of a claim, decision, or argument within the sentence. This task was complicated by the need to determine whether the judgment was relevant to the current case rather than previous or higher court judgments, which require a more circumstantial analysis than a judgment by itself. To deal with this, we developed expressions based on BERT, which were processed by a bidirectional LSTM network (Bi-LSTM) for the final groups, employing a similar approach as (Devlin et al., 2019). This integration leverages the contextual embeddings generated by BERT, enabling the model to capture complex dependencies within the text. The Bi-LSTM layer further refines these embeddings, enhancing the model’s ability to interpret and classify the data accurately. Another challenge was the excessive length of sentences, often exceeding the upper token limit of the BERT model when including contextual information. To overcome this, we adopted a strategy where the input of the sentences was generated in the form of a transformer, and the sliding window method was used to feed these inputs to the Bi-LSTM network, which allows the analysis of the standard without more. The objective of the model in the second phase was to identify specific features in each written sentence, which was implemented as a multiline classification problem. We analyzed and optimized transformer-based models on our data set of Spanish legal documents. Our analysis includes BERT-multilingual models and monolingual models such as BETO, as well as Spanish-specific BERT models. The main challenge is the imbalance of annotated and non-annotated sentences, with only 24% annotated. Instead of enhancing the text, which can be problematic in context-specific legal language, we adopted a naive down-sampling procedure to balance the datasets, although this did not significantly affect the F1 score. To optimize the model’s performance, we conducted 600 runs using the Optuna library, experimenting with a range of parameters for both stages of the classification process. This rigorous optimization aimed to fine-tune the model’s settings for the best possible performance on our unique dataset of Spanish legal texts.

Experimental results

To assess the efficiency of our model, we initially used the F1 score as a measure of performance. Because our data set is highly unbalanced, the tendency for F1 scores to inflate the results was evident, potentially leading to misleading interpretations. To combat this, we included a Kappa score, which is a more reliable agreement that is less affected by imbalanced information. Following the classification by Viera & Garrett (2005), we defined kappa scores ranging from 0.41 to 0.60 for moderate agreement, 0.61 to 0.80 for fair agreement, and 0.81 to 0.99 for nearly perfect agreement. The Kappa scores, accompanied by their standard errors, are detailed in the accompanying tables. Additionally, we confirmed the statistical significance of the F1 and Kappa scores using the chi-square test, which consistently demonstrated a strong correlation across all variables (p < 0.001). Our model undertook two primary tasks: the binary classification of sentences as either annotated or unannotated, and the multilabel classification of the annotated sentences. For the first task, we assessed inter-annotator agreement (IAA) across the entire dataset of 72,261 sentences, while the model’s performance was evaluated on a test set comprising 7,017 sentences. A sentence was considered annotated if it received a label from at least one of the annotators. The outcomes of these evaluations are summarized in the results in Table 4. The chi-square test was applied to the F1 and Kappa (K) values, revealing a significant correlation across all variables with p-values less than 0.001. The model undertook two primary tasks: firstly, distinguishing between annotated and non-annotated sentences within the full corpus of 72,261 sentences, and secondly, conducting multilabel classification on the annotated sentences, with performance assessed on a test subset of 7,017 sentences. A sentence was deemed annotated if tagged by at least one annotator, with detailed outcomes presented in the results section.

Table 4 Comparison of human and neural network annotation performance.

			Annotator 1				Neural network	
			NA (No. of annotations)	AN (Annotated number)				NA (No. of annotations)	AN (Annotated number)	
Annotator 2	NA		5,380 (74%)	3,530 (5%)	Annotators	NA	4,070 (58%)	1,079 (15%)	
	AN		5,442 (7%)	8,414 (11%)		AN	443 (6%)	1,425 (20%)	
	Precision	Recall	F1 Score	Kappa		Precision	Recall	F1 Score	Kappa	
	0.71	0.61	0.64	0.57		0.56	0.72	0.64	0.50	

Our analysis shows that 24% of sentences were annotated by human annotators, showing a Kappa score of 0.5782, which borders on substantial agreement. In contrast, the model identified a larger portion of sentences as annotated (35%), with a Kappa score of 0.4991 indicating moderate agreement. Despite similar F1 scores of around 0.65 for both human annotators and the model, the paths to this performance diverged. Human annotators achieved this through higher accuracy (P = 0.7044), whereas the model excelled in recall (R = 0.7628). This discrepancy suggests that differences in performance metrics might stem from factors beyond sample imbalance, such as the type of errors made (false positives and negatives), which could affect the reliability of extracted information from court rulings. Moving to the second task, a sentence was categorized based on the presence of at least one relevant label. The IAA was calculated using sentences jointly labeled by both annotators (N = 8,414), with agreement noted when both assigned the same category to a sentence. For the model’s performance evaluation, sentences from the test set (N = 3,328) were used, with an agreement between the model and annotators recognized if the model’s classification matched at least one annotator’s. The forthcoming results section details findings related to custody requests (RQ_JOIN) and decisions (DEC_JOIN), with “+” indicating individual custody and “–” signifying joint custody. Table 5 includes percentages of agreement, with the positive (+) and negative (–) signs indicating different subcategories within the request and decision categories. The accuracy, recall, F1 score, and K-index values are provided for each subcategory. The neural network’s performance is compared with that of the human annotators, indicated by Annotator 1, and Annotator 2, and when both annotators agree (Annotators+). The K-index values include standard errors, suggesting a degree of variance in the IAA and agreement between the network and humans.

Table 5 IAA and comparative analysis of model-human consensus on identified requests and decisions.

	Request (RQ_JOIN)	Decision (DEC_JOIN)	
	+	–	+	–	
Annotator 1	715	4	74	16	
	(8%)	(0%)	(1%)	(0%)	
Annotators +	137	14	14	9	
	(4%)	(0%)	(0%)	(1%)	
Annotator 2	0	1024	99	27	
	(0%)	(12%)	(1%)	(0%)	
Neural network	6440	223	35	2842	
	(77%)	(7%)	(1%)	(85%)	
Precision	0.96		0.9613		
Recall	0.971		0.9621		
F1 Score	0.971		0.9617		

Table 6 presents the IAA and the agreement between the neural network and the human annotators for various arguments within the court sentences. The IAA for requests and decisions showed very high agreement, nearing almost perfect agreement (K = 0.9243 for requests and K = 0.7924 for decisions). The neural network’s performance was also strong, with K values indicative of almost perfect to substantial agreement for requests (K = 0.8398) and decisions (K = 0.6343). For the arguments, the neural network exceeded the threshold of substantial agreement in all but one case. The IAA yielded similar K values, with the highest agreement observed for children’s opinions (CHILD_OPIN; K = 0.8005) and the availability of time and material means (PAR_RDNS; K = 0.8063). The lowest performance by the model was seen with the rootedness of the children (CHILD_ROOT; K = 0.6022), while the lowest IAA was for children’s circumstances (CHILD_CIRC; K = 0.5275).

Table 6 IAA and concordance between the model and humans regarding arguments identified in sentences.

Argument	Annotator 1 (N)	Annotator 1 (Argument)	Annotator 2 (N)	Annotator 2 (Argument)	Neural network (N)	Neural network (Argument)	Precision	Recall	F1 score	K (Index ± SE)	
PAR_RDNS	6,207 (74%)	297 (4%)	277 (3%)	1,633 (19%)			0.8550	0.8461	0.8505	0.8063 ± 0.0078	
CHILD_OPIN					6,207 (74%)	277 (3%)					
PSY_REP	7,835 (93%)	90 (1%)	114 (1%)	375 (4%)	2,177 (65%)	128 (4%)	0.8698	0.8358	0.8524	0.7888 ± 0.0117	
PAR_RELAT	7,951 (94%)	92 (1%)	90 (1%)	281 (3%)	3,195 (96%)	18 (1%)	0.7574	0.7534	0.7554	0.7441 ± 0.0182	
BEST_INT	5,974 (71%)	563 (7%)	299 (4%)	1,578 (19%)	3,184 (96%)	32 (1%)	0.8407	0.7370	0.7855	0.7186 ± 0.0090	
PAR_DED	7,435 (88%)	158 (2%)	270 (3%)	551 (7%)	2,064 (78%)	104 (3%)	0.6711	0.7772	0.7203	0.6924 ± 0.0140	
	3,004 (90%)	62 (2%)	78 (2%)	184 (4%)							

Table 7 presents the network model across three categories related to child custody cases: Children’s Opinions (CHILD_OPIN), Psychosocial Reports (PSY_REP), and Previous Dedication to the Children (PAR_DED). The percentage agreement among human annotators (Annotator 1 and Annotator 2) and evaluates the neural network’s performance. The results for the identification of arguments, as ordered by the Kappa value in IAA, indicate a range of consistency in labeling. The IAA for requests reflects nearly perfect agreement with a Kappa of 0.9243, suggesting an excellent level of concordance among human annotators in identifying these elements within the text. Similarly, the decisions also scored high with a Kappa of 0.7924, approaching the threshold of near-perfect agreement. The model’s performance, as assessed by the Kappa values, was impressive, particularly for requests where it achieved a Kappa of 0.8398, signifying almost perfect agreement. In the case of decisions, the model’s Kappa was 0.6343, indicative of substantial agreement. Regarding the arguments, the model’s performance surpassed the substantial agreement threshold for six of the arguments, with the highest model performance (K = 0.7371) for the availability of time and material means to care for the children (PAR_RDNS). The lowest model performance was observed in the rootedness of the children (CHILD_ROOT; K = 0.6022), which is marginally above the substantial agreement threshold. The IAA values for these categories were similarly high, with six arguments falling into the substantial agreement category. The highest Kappa values both for the model and IAA were observed for children’s opinions (CHILD_OPIN; K = 0.8005) and availability of time and material means (PAR_RDNS; K = 0.8063), respectively. The poorest performance by the model was found in the rootedness of the children (CHILD_ROOT; K = 0.6022), and the lowest IAA was observed for the children’s circumstances (CHILD_CIRC; K = 0.5275). The study aimed to ascertain whether judicial proceedings could be characterized solely by analyzing the legal grounds section of court rulings. Due to the inherent ambiguity of legal concepts and the absence of a gold standard for comparison, the IAA was used as a benchmark against the model’s performance. IAA was calculated across all labeled rulings (N = 2394), giving a broader measure of agreement, whereas the model’s performance was assessed against a subset (N = 595) in the test set. The findings showed that the annotators agreed with 76% of the rulings regarding the type of custody requested, with 31% of rulings identified as requesting individual custody and 45% requesting joint custody. The decisions were also largely in agreement, with a 63% consensus observed.

Table 7 IAA and agreement between network and humans on the arguments.

	Children’s opinions (CHILD_OPIN)	Psychosocial report (PSY_REP)	Previous dedication to the children (PAR_DED)	
Annotator 1	75%	75%	88%	
Annotator 2	5%	4%	3%	
Neural network	Precision	Recall	F1 Score	Kappa	Precision	Recall	F1 Score	Kappa	Precision	Recall	F1 Score	Kappa	
	0.77	0.76	0.77	0.70	0.75	0.63	0.65	0.59	0.74	0.55	0.62	0.60	
Annotator 1	83%	81%	66%	
Annotator 2	7%	4%	8%	
Neural network	Precision	Recall	F1 Score	Kappa	Precision	Recall	F1 Score	Kappa	Precision	Recall	F1 Score	Kappa	
	0.57	0.91	0.70	0.63	0.71	0.76	0.74	0.70	0.57	0.76	0.66	0.60	

Discussions

It is evident that the integration of neural network models in analyzing legal documents, particularly in the context of custody-related court rulings, offers substantial promise. The disparity in agreement rates between human annotators and the model underscores the complexity of legal language and the interpretative variability inherent in legal analysis.

Evaluating open-source algorithms

In our analysis, we assessed various open-source algorithms from the Hugging Face platform, employing multiple baseline models to ensure a thorough comparison. The configurations for these models were carefully selected to enhance performance and allow for comparative evaluation across different architectural frameworks. For the task of classifying sentences as annotated or non-annotated, we examined two main configuration groups. The first group included the combination of Term Frequency-Inverse Document Frequency (TF-iDF) with random forest and TF-iDF with Bi-LSTM as key methodologies. The TF-iDF configuration was set with minimum and maximum n-grams of 1 and 5, respectively, and was limited to 500 features. Table 8 summarizes the results from the initial stage of our analysis, focusing on the classification of sentences. The models were assessed based on their accuracy as determined by Optuna, a hyperparameter optimization framework, along with their accuracy and F1 scores on the test dataset. In optimizing the model, the Optuna library was used for hyperparameter tuning, involving 600 runs. The explored hyperparameters included learning rates from 1 × 10−5 to 1 × 10−2, batch sizes from 16 to 64, epochs from 5 to 20, dropout rates from 0.1 to 0.5, hidden units in Bi-LSTM layers from 50 to 200, maximum token lengths from 128 to 512 tokens, and optimizers like Adam, AdamW, and RMSprop. This comprehensive process identified a final configuration with a learning rate of 4 × 10−4, batch size of 32, 10 epochs, a dropout rate of 0.3, 100 hidden units, a maximum token length of 256 tokens, and the AdamW optimizer, achieving substantial agreement with human annotators.

Table 8 Efficacy of comparative analysis models.

Configuration approach	Optuna performance (%)	Accuracy in testing (%)	F1 outcome	
TFiDF Combined with RandomForest	88.9	78.4	0.65	
TFiDF Merged with Bi-LSTM	91.6	79.7	0.71	
Sentence Transformer and Bi-LSTM Fusion	82.0	80.4	0.71	
Sentence Transformer with Non-Recurrent NN	86.6	80.5	0.68	
Windowed Sentence Transformer Plus Bi-LSTM	78.6	78.1	0.69	

The performance of each model is evaluated using several metrics, including Label Ranking Average Precision (LRAP) across training, testing, and evaluation datasets, along with precision, recall, and F1 scores as shown in Table 9. For the classification task involving both annotated and non-annotated sentences, our study explored two main configurations. Initially, we utilized TF-iDF combined with random forest and TF-iDF with Bi-LSTM as our primary approaches. The TF-iDF settings were adjusted to include 1 to 5 n-grams and limit the feature set to 500, adhering to standard Spanish stop words and default document frequency parameters. The random forest model was set with 100 trees and default settings for the number of variables to split and the maximum depth of the trees, set at 10. The Bi-LSTM configuration was finalized with a single layer, running for five epochs, and included 100 hidden units, a batch size of 15, and a learning rate of 0.00082. The second configuration set focused on Sentence Transformer models paired with either a Bi-LSTM or a non-recurrent neural network (NN), along with an innovative approach that employed a Windowed Sentence Transformer in conjunction with Bi-LSTM. We used the “distiluse-base-multilingual-cased” model for the Sentence Transformer, maintaining the Bi-LSTM configuration as previously described. The non-recurrent NN setup, on the other hand, was structured with 15 epochs, a batch size of 31, a learning rate of 0.00082, and a dropout rate of 35%. The multilabel classification results, detailed in a subsequent table, utilized the F1 score to evaluate the model’s alignment with human annotators in labeling sentences.

Table 9 Performance evaluation of various NLP models.

Model type	Implementation	LRAP (Train)	LRAP (Test)	LRAP (Eval)	P	R	F	
BERT	BERT-base-multilingual-uncased	0.79	0.79	0.83	0.69	0.56	0.56	
BERT	BETO (Spanish BERT)	0.86	0.86	0.86	0.77	0.74	0.76	
DistilBERT	DistilBERT-base-multilingual-uncased	0.82	0.81	0.82	0.68	0.58	0.62	
XLM	XLM-mlm-enro-1024	0.81	0.81	0.81	0.62	0.60	0.61	
XLM-RoBERTa	XLM-RoBERTa-large	0.87	0.87	0.87	0.74	0.76	0.75	

Knowledge extraction

In this section, the nuances of knowledge extraction are explored, which delineates the F1 scores and Kappa indices reflecting the IAA and the alignment between the model and human annotators in categorizing various elements like requests, decisions, and arguments within sentences and broader court rulings. The findings reveal a modest level of concordance between the model’s classifications and the annotators’ judgments, evidenced by an F1 score of 0.6519 and a Kappa coefficient of 0.4991, which suggests a moderate level of agreement. Table 10 provides a comprehensive overview of the model’s performance in aligning with human annotators on labeling sentences and court rulings with requests, decisions, and arguments. The model is capable of filtering sentences within a court ruling that significantly contribute to its characterization from those that do not, essentially separating annotated from non-annotated sentences. Although the model’s concordance with annotators in this classification task isn’t exceptionally high, with an F1 score of 0.6519 and a Kappa index indicative of moderate agreement (K = 0.4991), it suggests room for improvement. The focus of the experiment was on how selecting pertinent sentences influences the classification of court rulings, given that the efficacy of this task impacts both sentence classification and overall ruling categorization. Ideally, challenges in accurately selecting relevant sentences would diminish the model’s effectiveness in categorizing court rulings compared to sentence classification. However, the model showed almost equivalent performance in both tasks, with minor differences in the average F1 scores (0.78 for sentences and 0.77 for court rulings) and Kappa indices (0.72 for sentences and 0.66 for court rulings). Yet, a deeper analysis of individual labels revealed a decrease in the correlation between IAA and model outcomes from 0.97 (F1) and 0.74 (K) in sentence classification to 0.65 (F1) and 0.29 (K) in court ruling classification, potentially due to the selection of relevant sentences. The model achieved an acceptable level of efficiency in selecting relevant sentences. The model’s proficiency is to identify the plaintiff’s request, the court’s decision, and predefined arguments within a court ruling. It was observed that the model could identify a request in 62% of court rulings and a decision in 48%, with better performance noted for requests (F1 = 0.87, K = 0.80) compared to decisions (F1 = 0.75, K = 0.57). This discrepancy might stem from the inherent challenges annotators face in locating explicit expressions within legal grounds that denote the request and decision, challenges that seem to amplify when applied to the neural network. Improvements in model performance could potentially be realized by broadening the analysis to include other sections of court rulings, particularly the facts section, which always outlines the request more explicitly than the legal grounds. This expansion could enhance request identification, though it’s important to note that the facts section in appeal court rulings also encompasses requests from earlier instances. Similarly, analyzing the verdict section, which explicitly states the decision, could refine the model’s ability to deduce decisions, thereby enhancing system performance in this aspect. The model shows promise in multilabel sentence classification with F1 scores ranging from 0.62 to 0.96 and Kappa indices from 0.60 to 0.84, suggesting substantial agreement for most labels, its performance in classifying court rulings is slightly lower, indicating room for further refinement and exploration in future research.

Table 10 A summary of the F1 score and Kappa index values for IAA and agreement between model and humans.

	F1 score (IAA)	F1 score (Model)	Difference	K index (IAA)	K index (Model)	Difference	
RQ_JOIN	0.97	0.96	−0.01	0.86	0.87	0.00	
DEC_JOIN	0.93	0.93	0.00	0.80	0.75	−0.03	
BEST_INT	0.72	0.72	0.00	0.60	0.78	0.10	
CHILD_CIRC	0.57	0.67	0.02	0.60	0.74	0.02	
CHILD_OPIN	0.79	0.81	0.06	0.76	0.69	−0.05	
CHILD_ROOT	0.60	0.62	0.00	0.62	0.66	0.01	
PAR_DED	0.64	0.70	0.00	0.61	0.69	0.02	
PAR_RDNS	0.85	0.85	0.00	0.85	0.93	0.08	
PAR_RELAT	0.79	0.80	0.01	0.83	0.78	−0.03	
PSY_REP	0.76	0.73	−0.03	0.67	0.76	0.02	

Table 10 shows a summary of the F1 Score and Kappa Index Values for IAA and agreement between model and humans.

The proposed neural network model leverages advanced transformer-based architectures like BERT and Bi-LSTM, distinguishing it from traditional machine learning approaches and earlier NLP models. The proposed model stands out by providing enhanced accuracy and context sensitivity through transformer-based neural networks, offering significant improvements over existing approaches (Aletras et al., 2016; Munoz Soro & Serrano-Cinca, 2021; Ashley & Walker, 2013; Katz, Bommarito & Blackman, 2017; Chalkidis, Androutsopoulos & Aletras, 2019) in both depth of analysis and prediction accuracy in the legal domain, shown in Table 11.

Table 11 Comparative analysis of predictive models for court decisions and legal text analysis.

Ref.	Model type	Transformers	Performance	Context sensitivity	Validation	NLP techniques	Legal text focus	
Proposed model	Neural Network (BERT + Bi-LSTM)	✓	High	Advanced	Experimental	Transformer-based NLP	Child custody decisions	
Ashley & Walker (2013)	Deep Learning (ANN)	x	High	Intermediate	Experimental	Neural network, shallow learning	U.S. Supreme Court rulings	
Katz, Bommarito & Blackman (2017)	Machine Learning (Logistic Regression)	x	Low	Basic	Formal (Benchmark)	Feature-based NLP	European Court of Human Rights cases	
Chalkidis, Androutsopoulos & Aletras (2019)	Neural Network (CNN)	x	High	Intermediate	Experimental	Convolutional neural networks	Legislative texts	
Raz (2002)	Hybrid (ML + Linguistic)	x	Moderate	Moderate	Formal (Testing)	Hybrid rule-based and ML	Asylum cases	

The novelty of the proposed model is an integration of BERT and Bi-LSTM. The traditional methods primarily were based on support vector machines (SVM) and logistic regression. Leveraging the deep contextual capabilities of BERT, the model improves performance in handling complex legal texts. High predictive accuracy surpasses many traditional approaches, which achieve only moderate to low-performance levels. In legal research, the model can automate the analysis of extensive judicial databases, enhancing the efficiency of identifying patterns and trends in past rulings, thus supporting more rigorous empirical studies on judicial behavior and the evolution of case law. For legal practitioners, the model serves as a decision-support tool by providing quick, data-driven insights into relevant case law, aiding in formulating legal strategies, drafting arguments, and making informed decisions. This capability improves the accuracy of legal advice and enhances case preparation by analyzing the intricacies of judicial decisions to predict potential outcomes based on historical data. In legal education, the model can be integrated into training programs, offering interactive tools for understanding judicial decisions and legal reasoning and simulating real-world scenarios to provide automated feedback on case analysis. This integration enhances learning experiences and improves analytical skills, preparing students for real-world legal challenges. However, ensuring data privacy, maintaining accuracy, and adapting the model to specific jurisdictions are essential considerations for integrating it into legal workflows and educational settings. These practical applications highlight the potential of the model to transform various aspects of the legal field, providing advanced tools for research, practice, and education while aligning technological advancements with the evolving needs of the legal community.

Conclusion

Our proposed model is a two-phase structured approach. In the initial phase isolation of pertinent sentences within the court ruling is performed. In the second phase, a congruence observed between the model’s outputs and the annotations provided by human reviewer’s points to a commendable level of accuracy in this preliminary task. The novelty of the proposed model is an integration of BERT and Bi-LSTM. The traditional methods are primarily based on SVM, and logistic regression. We have performed a comparison with the state-of-the-art models model’s based on F1 and Kappa metrics. Our proposed model’s performance is well as compared to the state-of-the-art model IAA.

Supplemental Information

Supplemental Information 1 Code

Additional Information and Declarations

Competing Interests

Author Contributions

Data Availability

The authors declare there are no competing interests.

Mohammad Abrar performed the experiments, analyzed the data, authored or reviewed drafts of the article, and approved the final draft.

Abdu Salam performed the experiments, authored or reviewed drafts of the article, and approved the final draft.

Faizan Ullah conceived and designed the experiments, authored or reviewed drafts of the article, and approved the final draft.

Muhammad Nadeem analyzed the data, performed the computation work, prepared figures and/or tables, authored or reviewed drafts of the article, and approved the final draft.

Hussain AlSalman performed the computation work, authored or reviewed drafts of the article, and approved the final draft.

Muaadh Mukred performed the computation work, prepared figures and/or tables, authored or reviewed drafts of the article, and approved the final draft.

Farhan Amin conceived and designed the experiments, prepared figures and/or tables, authored or reviewed drafts of the article, and approved the final draft.

The following information was supplied regarding data availability:

The data and code are available at Zenodo: Yeungnam University. (2024). Advanced neural network-based model for predicting court decisions on child custody [Data set]. Zenodo. https://doi.org/10.5281/zenodo.11657592.

The Judicial Documentation Centre (Cendoj) court rulings related to child custody cases from the Spanish judiciary are available at https://www.poderjudicial.es/portal/site/cgpj/menuitem.87fc234e64fd592b3305d5a7dc432ea0/?vgnextoid=cedd2bddeb0b6510VgnVCM1000006f48ac0aRCRDvgnextlocale=envgnextfmt=defaultlang_choosen=en. These rulings provided detailed information about the judgments, legal principles, arguments, and evidence presented in each case. The data was used to analyze patterns in judicial decisions and develop a model for predicting court outcomes in child custody disputes. The collected dataset was essential for training and validating the NLP and deep learning models used in the study.

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
