# Peer review of "Advanced neural network-based model for predicting court decisions on child custody"

_PeerJ Computer Science, doi:10.7717/peerj-cs.2293_

## Round 0.1 · original submission · Major Revisions

As also identified by the reviewer, transformer-based approaches are not new in such applications. So identify the major novelty and contributions of this work. Some rewriting efforts are needed for the abstract. Make it compact.
The reviewer has identified some mismatch in the reference citations. Please carefully revise the relevant literature.

Reviewer 1 ·

Basic reporting

- The language used in the manuscript is clear and professional.
- The introduction provides sufficient background and context for the research.
- The literature review is comprehensive and relevant, covering various NLP techniques and their applications in the legal domain.
- The structure of the manuscript is well-organized and follows a logical flow.
- The figures and tables are relevant and informative.
- The raw data and annotations are provided.

Experimental design

- The research question and the identified knowledge gap are well-defined and meaningful, addressing the prediction of court decisions on child custody cases.
- The methodology section is detailed and thorough, describing the annotation process, dataset compilation, and model development.
- The authors have employed rigorous techniques, including transformer-based models and neural networks, to address the research problem.
- The experimental setup, including the training, validation, and testing splits, is clearly described.

Validity of the findings

- The authors have provided a comprehensive discussion of the results, including comparisons with human annotators and inter-annotator agreement.
- The underlying data and annotations are robust, and the statistical analysis appears sound.
- The conclusions are well-stated and linked to the original research question, supported by the presented results.

Additional comments

There are some comments I would like to add that will improve quality of the manuscript.

- Dataset Limitations: While the dataset is commendable in size (3,047 court rulings), the authors acknowledge the limitation of manually downloading individual documents due to restricted bulk access. This constraint could potentially introduce selection bias and limit the generalizability of the findings. The authors should discuss this limitation more thoroughly and propose strategies to mitigate or address this issue in future research.
- Model Optimization and Hyperparameter Tuning: The authors mention conducting 600 runs using the Optuna library for optimizing model parameters. However, the specific details of the optimization process, including the range of hyperparameters explored and the rationale behind the chosen values, are not provided. A more detailed description of the model optimization process would strengthen the reproducibility and transparency of the research.
- Comparison with Existing Approaches: While the authors have provided a comprehensive literature review, a direct comparison of their proposed approach with existing techniques or methodologies for predicting court decisions or analyzing legal texts would be beneficial. This would help contextualize the novelty and performance of the proposed model, allowing readers to better understand its advantages and limitations compared to alternative approaches.
- Exploration of Advanced Language Models: The authors mention the potential of leveraging advanced language models like GPT-3 for enhanced performance. However, they do not provide any empirical evaluation or discussion of the potential benefits or challenges of using such models. Incorporating an exploratory analysis or discussion of advanced language models in the legal domain could further strengthen the contribution and impact of the research.
- Real-World Applications and Implications: While the authors briefly mention the potential applications of their work in legal research, practice, and education, a more detailed discussion of the practical implications and potential real-world applications would be valuable. This could include insights into how the proposed approach could be integrated into legal workflows, decision support systems, or educational tools, and the potential challenges or considerations associated with such applications.

Reviewer 2 ·

Basic reporting

1. The abstract need to be concise highlighting only the key motivation, contributions, methods and summary of results. In the current form the abstract is verbose. I would suggest the authors to consider rewriting the abstract.
2. The claim in the introduction section, line 74. The key difference between this study and previous ones is the incorporation of transformer-based model..., is not true in its current form. We have already some work in the child custody prediction based on transformers based models, please refer to [1].
3. In the key contributions highlighted by the authors I am unable to find relationship between the contribution 1. In this research, we introduce an innovative application of NLP... and reference [7]. S. C. Walpole, K. Smith, J. McElvaney, J. Taylor, S. Doe, and H. Tedd, "An investigation into hospital prescribers' knowledge and confidence to provide high-quality, sustainable respiratory care," Future Healthcare Journal, vol. 8, p. e272, 2021. The scope of the current work is completely different from the work that the authors have cited.
4. The references provided in the paper are not much relevant. For instance, the author have not reported any prior work on predicting court decisions on child custody; however there are many studies related to the target NLP application [1, 2, 3, 4].
5. The section 2.2. The Revolution of Transformer Models in NLP is not in the scope of the actual topic. Instead of discussing revolution of transformer models I would suggest to provide discussion about NLP/DL application in the child custody prediction.
6. The references provided for each of the transformer models in section 2.2 and table 2 are not correct. The author must provide the actual work that have proposed these models, i.e., BERT [5], RoBERTa [6], DistillBERT [7], and XLNet [8].

[1]. Juan, Yining, et al. "CustodiAI: A System for Predicting Child Custody Outcomes." Proceedings of the 13th International Joint Conference on Natural Language Processing and the 3rd Conference of the Asia-Pacific Chapter of the Association for Computational Linguistics: System Demonstrations. 2023.
[2]. Munoz Soro, Jose Felix, and Carlos Serrano-Cinca. "A model for predicting court decisions on child custody." PloS one 16.10 (2021): e0258993.
[3]. Emery, Robert E., Randy K. Otto, and William T. O'donohue. "A critical assessment of child custody evaluations: Limited science and a flawed system." Psychological Science in the Public Interest 6.1 (2005): 1-29.
[4]. Forslund, Tommie, Mårten Hammarlund, and Pehr Granqvist. "Admissibility of attachment theory, research and assessments in child custody decision‐making? Yes and No!." New Directions for Child and Adolescent Development 2021.180 (2021): 125-140.
[5]. Devlin, Jacob, et al. "Bert: Pre-training of deep bidirectional transformers for language understanding." arXiv preprint arXiv:1810.04805 (2018).
[6]. Liu, Yinhan, et al. "Roberta: A robustly optimized bert pretraining approach." arXiv preprint arXiv:1907.11692 (2019).
[7]. Sanh, Victor, et al. "DistilBERT, a distilled version of BERT: smaller, faster, cheaper and lighter." arXiv preprint arXiv:1910.01108 (2019).
[8]. Yang, Zhilin, et al. "Xlnet: Generalized autoregressive pretraining for language understanding." Advances in neural information processing systems 32 (2019).

Experimental design

1. The authors study the dataset thoroughly, to select the appropriate features. The dataset is in Spanish language; I believe the authors either used some sort of translator or some of the authors may be expert in Spanish language. I am curious how the authors were able to understand the dataset. Can you please elaborate this.
2. If the authors have used translator, still it would be difficult to fully understand complex legal language. If there is any English version of the dataset available, please provide reference to that.
3. Similarly, some information about the annotation process will also clear the whole process.

Validity of the findings

1. The authors have reported results for different ML and DL models for comparison; however there is no comparison provided with the prior studies e.g., [1] and [2] etc.
2. The symbols and measures used in the tables are not defined well e.g., in Table 6 NA, AN, P, R, F and K. I believe P, R, F and K are Precision, Recall, F1 score and Kappa, respectively; however it is not easy to judge about all these acronyms.

Additional comments

The authors conducted extensive experiments about predicting court decisions on child custody using ML and DL methods; however the overall presentation of the work is not sufficient for publication at this time. The paper extensive requires rewriting, to get published.

---

## Round 0.2 · Major Revisions

One of the reviewers has raised points that need to be addressed by the authors. Hence, I am recommended the paper for another round of review. The authors need to address the comments carefully respecting the efforts and time of the reviewers.

Reviewer 1 ·

Basic reporting

The authors have revised the introduction, backgrounds and motivation in this version.

Experimental design

The authors have improved and streamlined the experimental design. So, it looks a lot better than the previous version.

Validity of the findings

The additional agreement with human annotations and the addition of further statistical measures have helped to revise the paper. This gives a good validity of experimental findings as well.

Reviewer 2 ·

Basic reporting

I appreciate the authors efforts to update the manuscript, however I still have some major questions regarding the reporting of this manuscript.

1) First of all when responding to the reviewers comments the authors must specifically mention the section, page and line number of the modification made by the authors to address the reviewers comments. For instance,
a) In response to comment 2) the authors must specify the place where they have updated the manuscript.
b) In the response to my comment 3) Thank you for your feedback regarding the mismatch between our stated contributions and the referenced work. We recognize that reference [7] is not relevant to the scope of our research in NLP applications for legal text analysis. In response, we correct the reference to [39] accurately reflect the appropriate literature that aligns with our contributions. I cannot find reference [39] there.
c) In response to Comment no 3: Experimental design, To clarify, we have now included a detailed description in the methodology section of our paper. This section explains how the annotation process... which section?

2) The authors have reported different experiments but they have only reported their proposed model (BERT + Bi-LSTM) in table 11 where the authors only reported the comparison qualitatively. I am confused why this is so.

Experimental design

The authors have responded to my comments and made some updates.

Validity of the findings

The following problems have raised questions about the validity of the findings

1) In this work the authors claim to provide a novel integration of the BERT and Bi-LSTM (which is used for other applications) for predicting court decisions. The authors have also provided code and dataset used in this work, however, when I checked the code, surprisingly there were no traces of using BERT. If this is the actual code used in this work then the claims are not correct. If the authors have provided wrong code by mistake, I would like the authors to update the supplementary material.

2) The authors previously reported that they have used BERT, and now in the updated manuscript the authors claim to have used combination of BERT and BiLSTM, but there is no change in the results in the initial and updated manuscript.

Additional comments

Overall, this work need very major revision and the author need to check each and every portion.

---

## Round 0.3 · accepted · Accept

The authors have addressed the comments of the reviewers. The manuscript is in much better shape after the two rounds of reviews. No more review is needed as per my understanding.